# Microsatellite Dataset for Cultivar Discrimination in Spring Orchid (*Cymbidium goeringii*)

**DOI:** 10.3390/genes14081610

**Published:** 2023-08-11

**Authors:** Da Eun Nam, Min Ju Cha, Yae Dam Kim, Manisha Awasthi, Yuno Do, Sam-Geun Kong, Ki Wha Chung

**Affiliations:** Department of Biological Sciences and BK21 Team for Field-Oriented BioCore Human Resources Development, Kongju National University, 56 Gongjudaehak-ro, Gongju 32588, Republic of Koreadoy@kongju.ac.kr (Y.D.); kong@kongju.ac.kr (S.-G.K.)

**Keywords:** *Cymbidium goeringii*, cultivar authentication, microsatellite, simple sequence repeats (SSRs), spring orchid

## Abstract

*Cymbidium goeringii* Reichb. fil., locally known as the spring orchid in the Republic of Korea, is one of the most important and popular horticultural species in the family Orchidaceae. *C. goeringii* cultivars originated from plants with rare phenotypes in wild mountains where pine trees commonly grow. This study aimed to determine the cultivar-specific combined genotypes (CGs) of short sequence repeats (SSRs) by analyzing multiple samples per cultivar of *C. goeringii*. In this study, we collected more than 4000 samples from 67 cultivars and determined the genotypes of 12 SSRs. Based on the most frequent combined genotypes (CG1s), the average observed allele number and combined matching probability were 11.8 per marker and 3.118 × 10^−11^, respectively. Frequencies of the CG1 in 50 cultivars (*n* ≥ 10) ranged from 40.9% to 100.0%, with an average of 70.1%. Assuming that individuals with the CG1 are genuine in the corresponding cultivars, approximately 30% of *C. goeringii* on the farms and markets may be not genuine. The dendrogram of the phylogenetic tree and principal coordinate analysis largely divided the cultivars into three groups according to their countries of origin; however, the genetic distances were not great among the cultivars. In conclusion, this dataset of *C. goeringii* cultivar-specific SSR profiles could be used for ecogenetic studies and forensic authentication. This study suggests that genetic authentication should be introduced for the sale of expensive *C. goeringii* cultivars. We believe that this study will help establish a genetic method for the forensic authentication of *C. goeringii* cultivars.

## 1. Introduction

*Cymbidium goeringii* Reichb. fil., belonging to the family Orchidaceae is one of the most important and popular horticultural species in East Asia [1,2]. *C. goeringii* is locally known as the spring orchid in the Republic of Korea because it blooms in early spring. *C. goeringii* cultivars are classically divided into two types: the “flower-variant cultivar” showing characteristic phenotypes of flower color and shape and the “leaf-variant cultivar” showing characteristic phenotypes of leaf color or variegation pattern and shape (Figure 1). Plants showing characteristic phenotypes in both the leaf and flower are usually called “double-variant cultivar” [3]. In the Republic of Korea, thousands of *C. goeringii* cultivars have been registered by two orchid registration organizations: the Korea Orchid Registration Association (KORA; http://www.koreso.com/, accessed on 31 May 2023) and the Registration Committee of the Korea Orchid Union (RCKOU; http://www.kour.or.kr/, accessed on 31 May 2023).

*C. goeringii* cultivars with horticulturally rare phenotypes are actively traded commercially at high prices through direct sales between sellers and purchasers or online systems. Depending on the cultivar, prices vary widely from a few to hundreds of thousands USD. Orchid cultivators and purchasers frequently worry about non-genuine cultivars, in which plants belonging to different cultivars are provided instead of the real cultivars. 

*C. goeringii* cultivars originally grew in wild mountains, where pine trees commonly grow. Each spring orchid showing an unusual unique phenotype was selected by orchid collectors and registered as a specific cultivar. *Cymbidium* species are usually reproduced through self-pollination in wild fields [4,5]; however, the individual number of each cultivar was increased through asexual vegetative propagation as a method of separating the shoots of an individual plant. Theoretically, all individuals originating from a particular cultivar strain are genetically identical when mutations that have occurred after cultivar fixation are ignored. Therefore, forensic discrimination can be used to determine whether two *C. goeringii* plants originate from a common cultivar, using microsatellite profiling [3,6]. This application is similar to microsatellite genotyping and the comparison of genetic profiles between suspects and the evidence(s) collected at crime scenes.

When a new *C. goeringii* cultivar was registered, whole orchid photograph(s) and phenotypic features were provided to the orchid registration organizations (KORA or RCKOU); however, neither tissue sample nor genetic information was deposited. Therefore, tracking the exact origin of a particular cultivar is difficult. In addition, determining which of the plants with similar phenotypes but different genetic profiles are from the original cultivar, is challenging. In the case of the leaf-variant types, the origin could be predicted from the differences in leaf shape, color, and variegation patterns, but it is difficult to predict originality from the phenotypes of the flower-variant types. This is because blooming is observed only in spring and the subtle features of flower colors and shapes differ depending on the cultivation conditions. A recent study suggested that approximately 40% of the purchased spring orchids (with tagged names) among the 10 flower-variant cultivars in the Republic of Korea were not genuine cultivars [3]. 

Microsatellites, also known as simple sequence repeats (SSRs), were first reported in *C. goeringii* by Moe et al. [7], then several studies have examined microsatellite markers in molecular phylogenetic and horticultural studies of the genus *Cymbidium* [3,6,8,9,10,11,12] or the family Orchidaceae species [13,14,15,16,17,18,19]. In particular, Lee et al. suggested the potential usefulness of microsatellite combined genotypes as a forensic authentication tool for discriminating among *C. goeringii* cultivars [3]. Lee et al. determined microsatellite combined genotypes for ten Korean cultivars of *C. goeringii* [3]. Nam et al. analyzed the genetic kinship among cultivars using combined genotypes in four closely related cultivar groups [6].

This study aimed to determine the cultivar-specific combined genotypes (CGs) of SSRs by analyzing multiple samples per cultivar in *C. goeringii*. We collected more than 4000 samples from 67 cultivars and determined the genotypes of 12 SSRs. In addition, we analyzed the phylogenetic relationships among the examined cultivars.

## 2. Materials and Methods

### 2.1. Collection of C. goeringii Samples

In this study, 4048 *Cymbidium* samples were collected from 67 cultivars (Table 1). Among these, 61 cultivars (*n* = 3957) were originally collected from the mountains of the Republic of Korea, whereas six cultivars (*n* = 91) originated from Japan and China. Most samples were flower-variant cultivars; some were leaf- or double-variant cultivars. Five *C. goeringii* from Japan were collected for genotype comparisons and phylogenetic analyses. The Chinese cultivar, Hwanguhajeong belonging to *C. forestii* was sampled as anoutgroup. In addition to the established cultivar samples, 155 wild *C. goeringii* samples were collected from the Republic of Korea mountains. 

### 2.2. DNA Purification

Leaves or roots of the orchid plants were cut into 0.5–1 cm long fragments and disrupted using a TissueLyser II (Qiagen, Hilden, Germany). Genomic DNA was purified from the disrupted samples using the DNeasy Plant Mini Kit (Qiagen, Hilden, Germany). DNA concentration was determined using a NanoDrop 2000 (Thermo Fisher Scientific, Wilmington, NC, USA).

### 2.3. Multiplex PCR for 12 SSR Markers

The 12 SSR markers were amplified using two multiplex PCR systems. The multiplex system 1 included seven markers: CG415 (HQ842905.1), CG709 (HQ842922.1), CG722 (HQ842923.1), CG787 (HQ842926.1), CG1023 (HQ842937.1), CG1210 (HQ842946.1), and CG1281 (HQ842953.1) and system 2 included five markers: CG649 (HQ842919.1), CG1028 (HQ842938.1), CG1085 (HQ842942.1), CG1320 (HQ842954.1), and CG1400 (HQ842958.1), respectively. The forward primers were labeled with one of the four fluorescent dyes: VIC, FAM, PET, and NED. The primer sequences, concentrations, and repeat units of the microsatellite markers were as described by Nam et al. [6]. PCR mixture was prepared to 10 µL including 20 ng of genomic DNA, primer mixture, and 5 µL of 2 × PCR Master Solution of AccuPower Multiplex PCR Master Mix (Bioneer, Daejeon, Republic of Korea). The thermal cycling conditions were as follows: 10 min predenaturation at 95 °C, 35 cycles of 30 s at 95 °C, 30 s at 58 °C, and 1 min at 72 °C, and final extension of 30 min at 72 °C using a SimpliAmp Thermal Cycler (Applied Biosystems-Applied Biosystems, Foster City, CA, USA). The PCR products were resolved using a SeqStudio Genetic Analyzer (Thermo Fisher-Applied Biosystems, Foster City, CA, USA), and the genotypes were determined using the Gene Mapper program (NT, Ver. 6.1) (Thermo Fisher-Applied Biosystems, Foster City, CA, USA). The nomenclatures of the alleles were principally determined by the repeat numbers of the microsatellite motifs according to the recommendation of the DNA Commission of the International Society for Forensic Hemogenetics (ISFH) [20] and based on the information from Hyun et al. [9]. 

### 2.4. Phylogenetic and Sibling Analysis

Genetic distances among cultivars were determined from similarities in CGs using a GenAlEx (v6.5) [21]. A dendrogram of the phylogenetic tree was constructed from the dissimilarity matrix using the unweighted pair group method with arithmetic averages. In addition, the dissimilarity matrix was used to perform principal coordinate analysis (PCoA), which graphically represents the genetic relationships among *C. goeringii* cultivars. The sibling probability using CG profiles was determined using Bayes theory [22]. 

### 2.5. Statistical Analysis

Allele frequencies were calculated by counting the number of observed alleles in all examined samples. Reference allele frequencies were obtained from a wild Republic of Korean *C. goeringii* population (*n* = 155). The combined matching probability (CMP) for the CGs of the 12 SSRs was calculated using the PowerStatsV12 program (Promega, Madison, WI, USA). A simple program based on MATLAB (MathWorks, Natick, MA, USA) was designed to search for similar CG from a pool of several thousand CGs.

## 3. Results

### 3.1. Determination of Combined Genotypes

This study determined the CGs of 12 microsatellite markers for 61 Republic of Korean cultivars (3957 samples) and 6 Japanese and Chinese cultivars (91 samples). In principle, alleles were named by repeating the number of SSR core units, as described earlier [3,6,9]. Most of the markers were analyzed well but genotyping of some samples failed for markers such as CG649, CG787, CG1023, and CG1085. The failed markers, even when retested from the same plants, were not amplified, or were poorly amplified by PCR, suggesting the possibility of variations in the primer binding sites. 

The most and second most frequent CGs (indicated by CG1 and CG2, respectively), and their CMPs are shown in Appendix A. Based on CG1, the average observed allele number was 11.8 per marker with the highest number (16) in CG649 and CG709 and the lowest number (8) in CG1320. The average CMP was 3.118 × 10^−11^, ranging from 8.890 × 10^−10^ for Daehongbo to 4.496 × 10^−36^ for Cheongoksan. This powerful discrimination made it possible to determine whether a *C. goeringii* individual labeled as a certain cultivar was genuine.

### 3.2. Determination of Cultivar-Specific Combined Genotypes

Only limited information, such as short phenotypic descriptions and photographs, is available for registered orchid cultivars in the Republic of Korea. Therefore, it is difficult to determine which genotype is genuine for two samples with different genotype profiles. If the CGs were the same in the two samples, they might have originated from an identical plant; otherwise, they might have originated from different plants. For a specific cultivar, the most frequent CG (CG1) among the samples that belonged to the specific cultivar was determined as the representative genotype profile of the corresponding cultivar [3].

In this study, 50 Republic of Korean cultivars with a sample size of 10 or more were analyzed for the frequencies of CG1s (Table 2). The frequencies of the CG1s were higher than 50% in most examined cultivars, except for cultivars of Cheonhwangso (41.0%), Hallasan (47.5%), Sacheonwang (40.9%), and Geumsusan (45.5%). In particular, the second frequent CGs (CG1B) were observed at relatively high frequencies of 29.8% in Cheonhwangso and 30.0% in Hallasan. When the predominant and second frequent CGs were compared in the two cultivars (Cheonhwangso: CG1A: 11–19/14–15/17–34/12–12/24–24/13–13/17–17/15–15/12–12/16–16/12–12/20–20 vs. CG1B: 11–11/15–29.1/17–34/12–20/24–24/13–25/17–17/16–16/17–17/16–16/7–12/13.1–20, and Hallasan: CG1A: 11-15/15-15/17-17/17-17/24-24/13-13/12-12/14-14/17-17/10-16/10-10/13.1-13.1 vs. CG1B: 11-11/15-15/17-17/17-17/24-24/13-13/17-17/14-14/15-17/16-16/10-10/13.1-13.1; matching alleles are underlined), they were considerably similar to each other with high sibling probabilities of 71.016% and 99.997%, suggesting that the plant individuals with both pairs were genetically close relatives. The horticultural phenotypes of plants with CG1A and CG1B were indistinguishable between the two cultivars. Therefore, we concluded that the individuals with either CG1A or CG1B were genuine for the corresponding cultivars. The sum of the frequencies of CG1A and CG1B was 70.8% in Cheonhwangso and 77.5% in Hallasan.

Among the 50 cultivars (*n* ≥ 10), the frequency of the CG1 ranged from 40.9% (Sacheonwang) to 100.0% (Jinjusu) with an average of 70.1% (Figure 2). Assuming that individuals with the CG1 are genuine to the corresponding cultivars, approximately 30% of *C. goeringii* on the farms and markets may not be genuine. The average frequency of the second frequent combined genotype (CG2) was 7.3%. For the CG1s observed in 12 cultivars with fewer than 10 samples, they were considered insufficient for assigning representative SSR profiles for those cultivars (Appendix A). Therefore, we considered them as “probable CGs”.

### 3.3. Tracing Cultivar Origin for Samples Assumed to Be Non-Genuine

This study traced the actual cultivars in 29.9% of samples that were suggested to be non-genuine. Most of them did not belong to the 66 *C. goeringii* cultivars examined in this study; however, some were determined to be CG1 of other cultivars (Table 3).

In the cultivar Hwanggeumso, CG1s were detected in seven cultivars (Gwaneum, Youngchoonso, Namhaeso, Cheonhwangso, Geumhaso, Chanbo, and Hobakjeon). The second most frequent CG2 in Hwanggeumso was identical to the CG1 in Gwaneum (50 times observed). In addition, CG1s of Youngchoonso (seven times) and Namhaeso (five times) were frequently observed in the samples as Hwanggeumso. In the samples collected from the cultivar Gwaneum, five types of CGs were identical to the CG1s of different cultivars (Youngchoonso, Cheongeumso, Geumhaso, Hobakjeon, and Chanbo). Among these, Youngchoonso (thirty times) and Hobakjeon (five times) were frequently observed. CG1 in Gwaneum was frequently identified (six times) in the cultivar Cheonhwangso. In the samples of cultivars Cheonsoo and Cheonsa, the CG1 of the Japanese cultivar Chanbo was frequently observed ten times and five times, respectively. In the samples of the cultivar Agassi, CG1 of Jinjusu was identified seven times. In the samples of the cultivar Hongdaewang, CG1 of Jangdan was identified eight times. In the samples of cultivar Munsubong, the CG1 of Hyangsu was identified six times.

The CG1s of the Japanese cultivars Chanbo and Hobakjeon were observed quite frequently in the samples of several different cultivars: Chanbo in Hwanggeumso (three times), Gwaneum (one time), Cheonsoo (ten times), Cheonsa (five times), Hobakjeon in Hwanggeumso (one time), and Gwaneum (five times). For Republic of Korean cultivars, the CG1 of Youngchoonso was observed in the samples from three cultivars (Hwanggeumso, Gwaneum, and Cheongeumso); Gwaneum, in two cultivars (Hwanggeumso and Cheongeumso); and Geumhaso, in two cultivars (Hwanggeumso and Gwaneum). Interestingly, a Chinese cultivar Hwanguhajung which belongs to the *C. forestii* was once observed in the samples collected by the cultivar Wonmyoung.

### 3.4. Phylogenetic Analysis among Cultivars

A phylogenetic tree was prepared from the genetic distance matrix of the CG1s from the 67 cultivars (Figure 3). The dendrogram roughly divided the cultivars into three groups according to their origin. Cultivars of Republic of Korean origin were separated from Japanese cultivars (Hobakjeon, Changseongjihwa, Sumunsan, Jilbugeum, and Chanbo) and the *C. forestii* cultivar (Hwanguhajeong) originating from China.

The *C. goeringii* cultivars were also separated by the principal coordinate analysis (PCoA). The cumulative percentage variances of three principal components (cum %) accounted for 11.83%, 18.52%, and 25.05% (Figure 4). With features similar to those in the phylogenetic dendrogram, the PCoA roughly divided them into three groups according to their origin. However, the Republic of Korean cultivar, Hwansaeng was included in the cluster of Japanese cultivars, and some Republic of Korean cultivars, such as Cheonunso and Jinna, were located near the Japanese cultivars. In both the phylogenetic tree and PCoA, the cultivars were poorly separated by variant type.

## 4. Discussion

Each established cultivar of *C. goeringii* is asexually propagated by dividing the shoots of a single plant; therefore, individuals constituting a cultivar are genetically identical clones in principle. In this study, we determined the CGs of 12 SSR markers for more than 60 cultivars of *C. goeringii* using multiplex PCR and subsequent CG profiling. We previously reported CGs of 10 Republic of Korean flower-variant cultivars [3]. The present study determined the CGs and their phylogenetic distribution in large-scale samples, including more than four thousand samples. This study revealed a very strong power of discrimination and polymorphic information, with a powerful average CMP of 3.118 × 10^−11^; this implies that the possibility of the exact same genotype profile in two randomly chosen *C. goeringii* samples is less than 1 in 20 billion. Therefore, we believe that profiling the 12 SSR markers is an excellent forensically applicable method to discriminate cultivars (or individuals) and analyze phylogenetic relationships. In addition, the application of this method could be extended as the cultivar discrimination tool for other *Cymbidium* species such as *C. sinense*, *C. faberi*, *C. ensifolium*, and *C. kanran*.

The allele names of microsatellites are usually expressed as the relative repeat numbers of short sequence units in the given samples in the phylogenetic and pedigree construction or linkage analyses. This study named alleles based on the absolute values of repeat numbers, similar to that used in forensic genetics and criminal investigations [20]. Therefore, the *C. goeringii* SSR dataset from this study will enable comparative research by ensuring data compatibility among research groups; in addition, it will serve as a standardized reference SSR database. For all examined cultivars, the most common SSR profiles (CG1) and the second most frequent profiles (CG2) with allele names indicated by repeat numbers are presented in Appendix A.

The most frequent CGs (CG1s) were observed in 2664 samples from the 50 cultivars with 10 or more samples (*n* = 3923). The mean of the obtained CG1 ratio for each cultivar showed a similar value of 70.1%. If we assume that only samples with CG1 are genuine to the corresponding cultivars, approximately 30% of the spring orchid cultivars in the Republic of Korean markets and farms may not be genuine. 

Non-genuine sales of spring orchids usually involve cultivars showing similar horticultural phenotypes but different prices. The cultivar Hwanggeumso is subject to frequent non-genuine sales. The cultivar Hwanggeumso has a superior phenotype among the class of yellow flowers with a non-anthocyanin white lip, and its price has remained steady at approximately USD 5000. The most frequently observed non-genuine cultivar of Hwanggeumso is Granum. The price of Gwaneum, which has yellow flowers with a non-anthocyanin white lip, is approximately USD 1000. In addition, Youngchoonso is frequently observed as a non-genuine cultivar of Hwanggeumso and Gwaneum. Youngchoonso belongs to the class of yellow flowers with non-anthocyanin white lips. However, its price is approximately USD 300, which is much cheaper than that of Hwanggeumso and Gwaneum. In particular, two Japanese cultivars (Chanbo and Hobakjeon) are sold as Republic of Korean cultivars. These two Japanese cultivars are far cheaper than their Republic of Korean counterparts, such as Hwanggeumso, Gwaneum, Cheonsoo, and Cheonsa.

The frequencies of the second frequent CGs were generally low, below 15%; however, they were high in two cultivars Cheonhwangso (29.8%) and Hallasan (30.0%). In the cultivars of Cheonhwangso and Hallasan, the most common CG (CG1A) and the second most common CG (CG1B) showed similar profiles with high sibling probabilities. In addition, the horticultural phenotypes of both plants having these different CGs were so similar that they are indistinguishable. Therefore, these were a pair of genetically close sister cultivars. Orchid cultivars collected from nearby wild locations that exhibit similar phenotypes are traditionally called sister cultivars. Nam et al. determined the genetic kinship between four groups of closely related sister cultivars of *C. goeringii* [6]. If individual plants with CG1A or CG1B in Cheonhwangso and Hallasan were sister cultivars, it would be difficult to determine which plant was the original. This gives rise to the question of whether plants with both CG1A and CG1B should be considered genuine cultivars of Cheonhwangso or Hallasan.

The phylogenetic tree based on SSR genotypes divided the cultivars into three groups according to their country of origin. The 61 Republic of Korea-origin cultivars were largely separated from the five Japan-origin cultivars and the China-origin *C. forestii* cultivar in the phylogenetic tree. These results were consistent with those of a previous study [5]. However, the genetic distances among the *C. goeringii* cultivars seemed to be close; even *C. forestii* from China was not far from the cultivars of *C. goeringii*. In the PCoA, *C. forestii* was located between the Republic of Korean and Japanese cultivar groups but was closer to the Japanese group. Several Republic of Korean cultivars were located around a cluster of Japanese cultivars. 

The complete nuclear genome sequence of *C. goeringii* is not available; however, sequences of the full chloroplast genome (cpDNA) of *C. goeringii* and a hybrid of *C. goringii* and *C. sinense* are available [23,24]. In addition, the recent RNA sequencing and transcriptomic analyses have suggested molecular mechanisms for phenotyping leaf color, floral patterning, and scent [25,26,27,28]. If genetic information on cpDNA variation and RNA expression is added to the dataset of SSR profile in future studies, a much more reliable cultivar discrimination and phylogenetic characterization can be performed in the *Cymbidium* species.

## 5. Conclusions

We examined 61 Republic of Korean cultivars of *C. goeringii* by genotyping 12 SSR markers for forensic genetic discrimination. The established dataset for *C. goeringii* cultivar-specific SSR profiles could be used for ecogenetic studies and forensic authentication. This study revealed that almost 30% of *C. goeringii* in the market may not be genuine. Therefore, we suggest that genetic authentication should be introduced for the sale of expensive *C. goeringii* cultivars. In addition, we suggest the preparation of guidelines for the DNA deposition and profiling of SSR genotypes in newly registered *C. goeringii* cultivars. We believe that this study will help establish a genetic method for the forensic authentication and phylogenetic analysis of *C. goeringii* cultivars.

## Figures and Tables

**Figure 1 genes-14-01610-f001:**
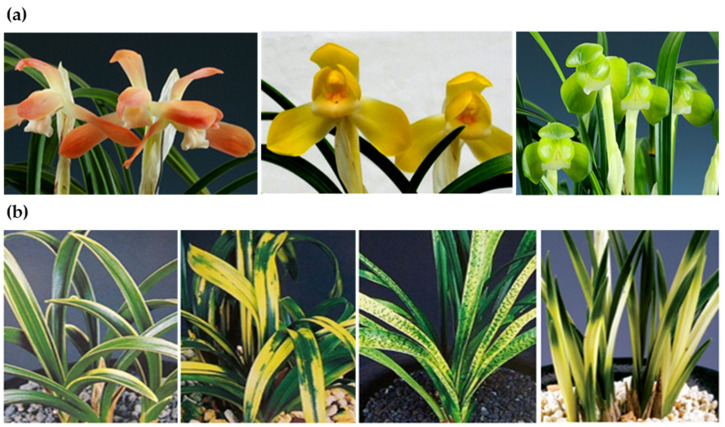
Cultivars of *Cymbidium goeringii*. (**a**) Flower-variant types. (**b**) Leaf-variant types.

**Figure 2 genes-14-01610-f002:**
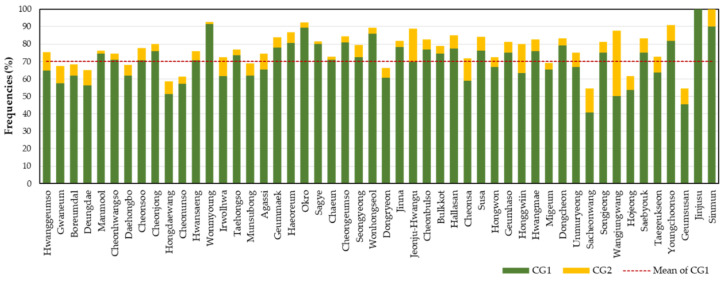
Frequencies of combined genotypes of 12 SSR markers in 50 *C. goeringii* cultivars with samples of 10 or more (CG1: most frequent combined genotypes, CG2: second frequent combined genotypes). The red dotted line represents the mean frequency of CG1.

**Figure 3 genes-14-01610-f003:**
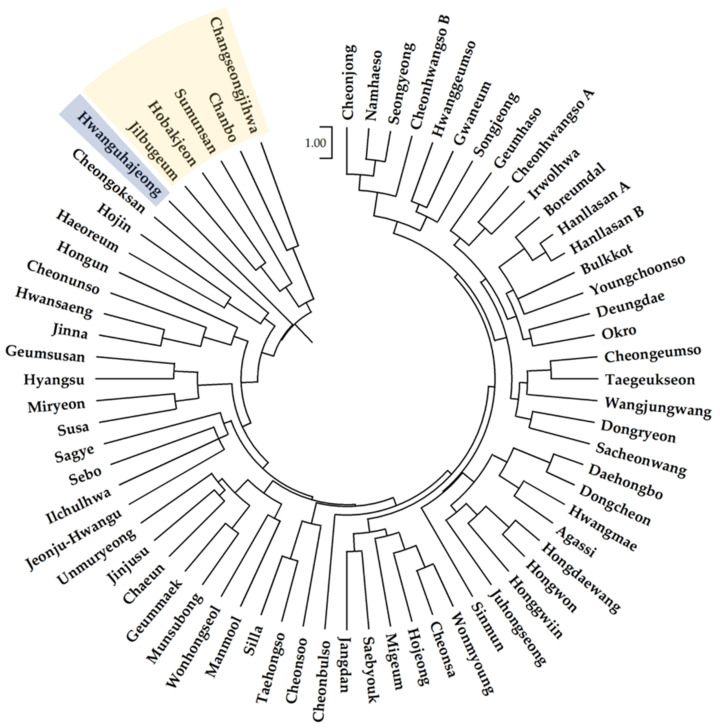
Phylogenetic tree of 66 cultivars in *C. goeringii* based on the genetic distance measured from genotypes of 12 SSR loci using the unweighted pair group method with arithmetic average (UPGMA) as the cluster method. The five cultivars within a yellow box are Japan-origin *C. goeringii*, while the Hwanguhajeong within a blue box is China-origin *C. forestii*.

**Figure 4 genes-14-01610-f004:**
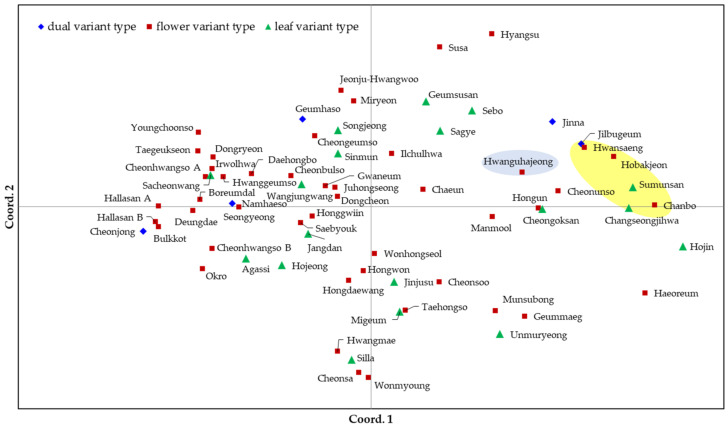
Principal coordinate analyses (PCoA) of 67 cultivars of *C. goeringii* or *C. forestii*. The five cultivars within the yellow ellipse are Japan-origin *C. goeringii*, while Hwanguhajeong within the blue ellipse is a China-origin *C. forestii*. The cumulative variability captured by the three principal component axes is 25.1% (PC1: 11.8%; PC2: 6.7%; PC3: 6.5%).

**Table 1 genes-14-01610-t001:** List of *Cymbidium goeringii* cultivars.

Cultivar Name	Type ^1^	Main Phenotype	Sample Number
Cultivars originated from Republic of Korea			
Hwanggeumso	F	Yellow flower with non-anthocyanin lip	468
Gwaneum	F	Yellow flower with non-anthocyanin lip	301
Boreumdal	F	Yellow flower with non-anthocyanin lip	250
Deungdae	F	Orange–yellow flower with non-anthocyanin lip	194
Manmool	F	Orange–yellow flower	184
Cheonhwangso	F	Yellow flower with non-anthocyanin lip	161
Daehongbo	F	Red flower	144
Cheonsoo	F	Orange–yellow flower with non-anthocyanin lip	143
Cheonjong	D	Short leaf and round flower both with yellow variegation	125
Hongdaewang	F	Red flower	109
Cheonunso	F	Flower with mixed colors and non-anthocyanin lip	98
Hwansaeng	F	Yellow flower with non-anthocyanin lip	95
Wonmyoung	F	Round yellow flower	93
Irwolhwa	F	Small round flower with non-anthocyanin lip	91
Taehongso	F	Orange–yellow flower with non-anthocyanin lip	91
Munsubong	F	Flower with mixed colors	84
Agassi	L	Leaf with narrow yellow stripes	78
Geummaek	F	Small round yellow flower	68
Haeoreum	F	Red flower	67
Okro	F	Flower with mixed colors and orange bordering	66
Sagye	L	Short leaf with yellow variegation	65
Chaeun	F	Orange–yellow flower with non-anthocyanin lip	62
Cheongeumso	F	Yellow flower with non-anthocyanin lip	58
Seongyeong	F	Leaf with yellow stripes and flower with non-anthocyanin lip	58
Wonhongseol	F	Purple small round flower	57
Dongryeon	F	Leaf with short yellow stripes and small round flower	56
Jinna	D	Leaf and flower with yellow variegation	55
Jeonju-Hwangu	F	Yellow flower	53
Cheonbulso	F	Red flower with non-anthocyanin lip	52
Bulkkot	F	Red flower	47
Hallasan	F	Small round flower	40
Cheonsa	F	Orange–yellow flower with non-anthocyanin lip	39
Susa	F	Red flower	38
Hongwon	F	Round red flower	36
Geumhaso	D	Leaf and flower with yellow stripe	32
Honggwiin	F	Orange–yellow flower with non-anthocyanin lip	30
Hwangmae	F	Pale yellow round flower	29
Migeum	L	Leaf with short multiple yellow stripes and spots	26
Dongcheon	F	Orange–yellow flower with non-anthocyanin lip	24
Unmuryeong	L	Overall yellowish leaf	24
Sacheonwang	L	Leaf with yellow stripes	22
Songjeong	L	Leaf with narrow yellow stripes	16
Wangjungwang	L	Short leaf with yellow stripes	16
Hojeong	L	Short leaf with yellow stripes	13
Saebyeok	F	Red flower with non-anthocyanin lip	12
Taegeukseon	F	Flower with yellow and green mixed colors	11
Youngchoonso	F	Yellow flower with non-anthocyanin lip	11
Geumsusan	L	Overall yellowish leaf	11
Jinjusu	L	Leaf with yellow stripes	10
Sinmun	L	Leaf with yellow stripes	10
Sebo	L	Leaf with yellow stripes	6
Silla	L	Short leaf with yellow bordering	5
Ilchulhwa	F	Small round flower	4
Cheongoksan	L	Leaf with multiple yellow spots	3
Hongun	F	Red flower	3
Jangdan	F	Red flower	3
Miryeon	F	Round flower with pale anthocyanin lip	3
Hojin	L	Leaf with yellow variegation	2
Hyangsu	F	Green flower with yellow bordering	2
Namhaeso	D	Leaf and flower with white bordering	2
Juhongseong	F	Flower with mixed colors and orange bordering	1
Cultivars originated from Japan			
Chanbo	F	Yellow flower with non-anthocyanin lip	40
Jilbugeum	D	Leaf and flower with multiple yellow stripes and spots	14
Changseongjihwa	L	Leaf showing leopard print	13
Hobakjeon	F	Orange–yellow flower with non-anthocyanin lip	11
Sumunsan	L	Leaf with yellowish variegation	8
Cultivar originated from China			
Hwanguhajeong ^2^	F	Floral fragrance	5
Total	-	-	4048

^1^ F: flower-variant type, L: leaf-variant type, D: dual-variant type. ^2^ It belongs to *C. forestii*.

**Table 2 genes-14-01610-t002:** Combined genotypes of 12 SSRs observed in *C. goeringii* cultivars with samples of 10 or more.

Cultivars	CG1%	Genotypes of SSR Markers ^1^											
		CG415	CG649	CG709	CG722	CG787	CG1023	CG1028	CG1085	CG1210	CG1281	CG1320	CG1400
Hwanggeumso	64.7	11–11	15–30.1	17–18	12–20	24–29	13–13	15–17	14–15	29–29	16–16	10–13	13.1–20
Gwaneum	57.4	11–13	21–30.1	18–18	12–20	24–24	13–13	15–15	14–19	15–29	16–16	10–13	13.1–20
Boreumdal	62.0	11–20	15–15	17–25	17–17	24–24	13–13	6–7	14–14	6–6	16–16	10–10	15–20
Deungdae	56.2	11–15	16–16	17–17	12–12	24–24	13–13	7–19	14–38	15–29	16–16	12–12	20–21
Manmool	74.5	13–13	15–29.1	23–23	12–17	18–18	13–20	15–17	19–19	17–17	9–9	12–12	20–20
Cheonhwangso	41.0 (A)	11–19	14–15	17–34	12–12	24–24	13–13	17–17	15–15	12–12	16–16	12–12	20–20
	29.8 (B)	11–11	15–29.1	17–34	12–20	24–24	13–25	17–17	16–16	17–17	16–16	7–12	13.1–20
Daehongbo	61.8	13–22	15–15	17–18	20–20	24–24	13–13	17–17	Null	15–15	16–16	7–7	19–19
Cheonsoo	70.6	11–13	14–15	18–18	17–20	24–24	20–20	17–17	14–14	15–15	10–16	7–12	20–20
Cheonjong	76.0	11–11	15–19	17–17	12–20	24–24	13–13	6–30	14–14	15–15	9–9	10–15	13.1–20
Hongdaewang	51.4	13–13	15–33.1	17–22	12–20	24–24	19–19	30–30	14–14	6–15	16–16	6–7	19–20
Cheonunso	57.1	13–13	15–15	18–18	30–30	Null	26–26	17–17	14–14	15–15	16–16	7–7	10.1–13.1
Hwansaeng	70.5	13–13	26.1–30.1	18–18	11–11	Null	13–13	6–12	14–15	17–29	16–16	7–9	20–20
Wonmyoung	91.4	11–13	15–15	22–22	11–12	24–24	13–20	17–17	15–15	15–15	16–16	12–12	13.1–13.1
Irwolhwa	61.5	11–19	15–15	17–34	12–17	22–24	13–13	15–15	14–14	15–33	9–9	7–7	19–19
Taehongso	73.6	11–13	15–20	17–17	12–20	22–22	20–20	6–6	14–14	15–28	9–16	12–12	15–15
Munsubong	61.9	13–13	25.1–29.1	17–22	12–12	22–24	20–20	15–17	14–19	12–15	16–16	7–12	13.1–18
Agassi	65.4	13–22	15–15	17–17	12–20	24–24	13–20	15–17	16–16	6–15	10–17	7–12	19–19
Geummaek	77.9	13–13	15–29.1	17–22	11–12	22–25	13–20	6–12	19–19	12–15	10–16	10–12	19–20
Haeoreum	80.6	11–13	14–14	25–25	11–11	25–25	20–20	17–17	14–14	17–17	9–9	7–7	13.1–13.1
Okro	89.4	11–13	15–20	17–17	12–22	24–24	13–13	7–17	14–15	12–15	16–16	6–12	13.1–13.1
Sagye	80.0	13–15	15–26.1	18–26	12–12	16–22	13–25	17–30	14–15	6–17	9–16	7–12	15–15
Chaeun	71.0	13–13	29.1–29.1	16–16	12–12	22–24	13–13	15–15	14–14	17–17	9–16	7–13	13.1–20
Cheongeumso	81.0	13–19	15–29.1	17–25	12–17	24–25	13–13	17–29	14–24	15–15	16–16	7–7	13.1–19
Seongyeong	72.4	11–11	15–29.1	13–17	12–20	17–24	13–13	17–17	14–14	15–29	9–16	6–6	20–20
Wonhongseol	86.0	13–13	16–34.1	13–17	12–17	24–24	13–20	7–17	14–19	11–17	9–9	7–7	13.1–20
Dongryeon	60.7	15–15	15–15	17–25	12–20	22–24	13–13	15–15	14–14	15–15	16–16	12–12	20–20
Jinna	78.2	10–13	14–29.1	18–18	11–17	Null	13–13	7–7	14–14	15–15	16–16	7–7	19–19
Jeonju-Hwangu	69.8	13–15	15–15	34–34	12–20	Null	13–13	7–24	14–14	12–12	16–16	7–7	15–19
Cheonbulso	76.9	11–19	23–26.1	25–34	20–20	24–24	13–13	30–30	14–14	15–15	16–16	12–12	20–20
Bulkkot	74.5	10–11	15–15	17–17	12–12	24–24	13–13	17–17	Null	12–13	10–16	12–12	19–20
Hallasan	47.5 (A)	11–15	15–15	17–17	17–17	24–24	13–13	12–12	14–14	17–17	10–16	10–10	13.1–13.1
	30.0 (B)	11–11	15–15	17–17	17–17	24–24	13–13	17–17	14–14	15–17	16–16	10–10	13.1–13.1
Cheonsa	59.0	11–13	15–25.1	17–18	11–12	24–24	13–20	17–17	14–14	6–15	9–16	7–12	19–20
Susa	76.3	13–22	14–20	13–22	12–17	15–22	13–13	17–17	14–15	12–15	10–16	7–7	13.1–19
Hongwon	66.7	13–13	15–15	19–22	12–20	24–24	19–19	30–30	14–14	15–29	16–16	7–7	19–19
Geumhaso	75.0	11–19	14–34.1	18–26	12–17	24–24	13–13	6–17	14–19	15–17	16–16	7–13	13.1–20
Honggwiin	63.3	13–13	15–34.1	22–22	12–20	24–24	13–13	7–12	14–14	17–17	16–16	10–10	13.1–13.1
Hwangmae	75.9	13–13	15–15	17–17	11–22	24–24	13–20	17–30	13–14	15–15	16–16	7–7	20–23.1
Migeum	65.4	11–19	15–15	22–25	11–11	19–24	13–20	6–6	14–14	17–17	10–10	7–7	19–20
Dongcheon	79.1	13–13	15–15	18–18	20–20	24–24	13–13	12–17	14–14	15–15	16–16	7–7	13.1–13.1
Unmuryeong	66.7	13–13	15–29.1	16–17	11–11	17–24	13–20	12–12	14–14	12–12	16–16	10–10	19–19
Sacheonwang	40.9	15–15	15–15	13–13	12–12	24–24	13–13	6–7	14–14	15–15	16–16	7–13	20–20
Songjeong	75.0	11–12	19–30.1	17–18	12–20	17–18	13–13	17–17	24–25	29–30	16–16	13–13	13.1–20
Wangjungwang	56.3	13–15	15–15	18–25	12–22	24–24	13–29	7–17	15–19	15–17	17–17	10–13	13.1–20
Hojeong	53.8	11–11	15–23	13–22	11–12	24–24	13–13	16–16	14–22	15–15	9–9	12–13	13.1–20
Saebyeok	75.0	11–15	15–32.1	17–18	11–22	22–24	13–13	30–30	14–14	15–15	16–16	12–12	19–20
Taegeukseon	63.6	13–15	15–15	17–25	12–17	24–24	13–13	17–17	14–14	15–15	16–16	6–7	19–20
Youngchoonso	81.8	15–15	14–15	17–17	12–17	24–24	13–13	17–17	15–15	15–17	8–8	7–12	13.1–13.1
Geumsusan	45.5	13–22	14–14	17–17	17–20	25–25	13–13	17–17	15–15	15–17	9–9	7–7	19–19
Jinjusu	100.0	12–13	15–29.1	12–17	11–12	22–24	13–13	15–17	14–16	17–27	9–16	7–10	19–20
Sinmun	90.0	13–15	15–29.1	18–18	17–23	24–24	13–13	6–17	14–14	15–17	16–16	7–7	17–20

^1^ Allele were named by the repeat numbers of SSR motif according to the guideline of the ISFH. Abbreviations: CG: combined genotype, CMP: combined matching probability, SSR: simple sequence repeats.

**Table 3 genes-14-01610-t003:** Tracing of miss-matched *C. goeringii* cultivars (*n* ≥ 10).

Cultivar Names Mentioned by Suppliers	Genetically Matching Cultivars ^1^
Hwanggeumso (*n* = 468)	Gwaneum (50), Youngchoonso (7), Namhaeso (5), Cheonhwangso (1), Geumhaso (1), Chanbo (3) ^2^, Hobakjeon (1) ^2^
Gwaneum (*n* = 301)	Youngchoonso (30), Cheongeumso (2), Geumhaso (1), Hobakjeon (5) ^2^, Chanbo (1) ^2^
Boreumdal (*n* = 250)	Saebyeok (1)
Cheonhwangso (*n* = 161)	Gwaneum (6), Youngchoonso (1)
Daehongbo (*n* = 144)	Wonmyoung (1)
Cheonsoo (*n* = 143)	Chanbo (10) ^2^
Cheonjong (*n* = 125)	Sebo (3), Hwansaeng (1)
Hongdaewang (*n* = 109)	Jangdan (8)
Cheonunso (*n* = 98)	Taegeukseon (4)
Hwansaeng (*n* = 95)	Deungdae (1)
Wonmyoung (*n* = 93)	Hwanguhajeong (1) ^3^
Irwolhwa (*n* = 91)	Miryeon (1), Cheonjong (1)
Munsubong (*n* = 84)	Hyangsu (6)
Agassi (*n* = 78)	Jinjusu (7), Sinmun (3), Sebo (1)
Geummaek(*n* = 68)	Ilchulhwa (4)
Chaeun (*n* = 62)	Cheonsa (1)
Cheongeumso (*n* = 58)	Wonhongseol (1)
Cheonbulso (*n* = 52)	Hwanggeumso (1), Sacheonwang (1)
Bulkkot (*n* = 47)	Juhongseong (2)
Cheonsa (*n* = 39)	Chaeun (1), Chanbo (5) ^2^
Hongwon (*n* = 36)	Hongun (1)
Sacheonwang (*n* = 22)	Sinmun (3)
Songjeong (*n* = 16)	Sebo (1)
Taegeukseon (*n* = 11)	Songjeong (1)

^1^ Numbers in parentheses are observed numbers. ^2^ Japan-origin cultivar. ^3^ China-origin cultivar of *C. forestii*.

## Data Availability

Not applicable.

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
