# Peer review of "Microsatellite Dataset for Cultivar Discrimination in Spring Orchid (Cymbidium goeringii)"

_genes, 2023, doi:10.3390/genes14081610_

Round 1
Reviewer 1 Report
The ms is basically well written and the data are properly presented. I would think that the ms could be acceptable for publication in Genes after the five minor modifications specified as below.
L86: Please check if the correct sample number is 4,048 instead of 4,046.
L154: Please change "sequences" to "binding site".
L160: Please change "labelled" to "labled".
L165: Please change "identical plants" to "an identical plant".
L255 and L256: Please change "box" to "ellipse".
The ms is basically well written and the data are properly presented. I would think that the ms could be acceptable for publication in Genes after the five minor modifications specified as below.
L86: Please check if the correct sample number is 4,048 instead of 4,046.
L154: Please change "sequences" to "binding site".
L160: Please change "labelled" to "labled".
L165: Please change "identical plants" to "an identical plant".
L255 and L256: Please change "box" to "ellipse".
Author Response
Thank you very much for your valuable comments. We carefully edited the manuscript point by point according to your comments.
L86: Please check if the correct sample number is 4,048 instead of 4,046.
[Answer] According to the comment, sample number was corrected to 4,048.
L154: Please change "sequences" to "binding site".
[Answer] As the comment, "sequences" was changed to "binding sites”.
L160: Please change "labelled" to "labled".
[Answer] As the comment, "labelled" was changed to "labeled".
L165: Please change "identical plants" to "an identical plant".
[Answer] As the comment, "identical plants" was changed to "an identical plant".
L255 and L256: Please change "box" to "ellipse".
[Answer] As the comment, "box" was changed to "ellipse".
Reviewer 2 Report
The authors have presented the results of very interesting research with high potential for practical use. I have left several comments in the file attached. Aside from that, I would suggest the authors expand the discussion to other species with similar problems of confirming the origin/certification, etc., and explore if genotyping has previously been used for solving this particular problem. Some additional statistical analyses would be beneficial (see attached comments).

The language is good for the most part. Here and there I found usage of inappropriate terms when describing the results. In my opinion, the manuscript would benefit from proofreading by a native speaker.
Author Response
Thank you very much for your valuable comments. We carefully reflected your correction indicated in the attached file. In addition, we added following sentence in “Discussion”: In addition, the application of this method could be extended as the cultivar discrimina-tion tools for the other Cymbidium species such as C. sinense, C. faberi, C. ensifolium, and C. kanran
The language is good for the most part. Here and there I found usage of inappropriate terms when describing the results. In my opinion, the manuscript would benefit from proofreading by a native speaker.
[Answer] Thank you very much for your valuable comments. The manuscript was edited by an English editing organ (Editage).
Reviewer 3 Report
The MS undertakes the forensic authentication and separation of cultivars of a valuable horticultural orchid species, Cymbidium goeringii. It is difficult to separate the individuals of recorded cultivars based on their small phenological differences. The MS therefore aims to develop a genetic method, which it successfully implements by determining and comparing the genotypes of 12 SSRs. It should be noted that by examining more than 4,000 samples of the 67 variants, an SSR database was also created that also serves as a reference SSR database. This can form the basis of further research (323-325), and can also play a role in the registration of new cultivars (332-333).
Some bugs to fix:
Ad 70-73 The sentence should be reworded due to citation 9 (…, then…)
Ad 169 It is recommended to expand the explanatory / accompanying text of Table 2.
Ad 220-221 Please implement the accompanying text of Table 3.
Ad 358 Correctly: …seedlings….
Scientific Latin names are not highlighted / written in italics in the References chapter several times: 353, 357, 388-389
The use of the English language in the MS corresponds to the quality expected in a scientific paper.
Author Response
Thank you very much for your valuable comments. We carefully edited the manuscript point by point according to your comments.
Some bugs to fix:
Ad 70-73 The sentence should be reworded due to citation 9 (…, then…)
[Answer] As the comment, reference order was changed (7>8, 8>9, 9>7), and the sentence was edited as followed: … were first reported in C. goeringii by Moe et al [7], then several studies have examined microsatellite markers in molecular genetic studies of Cymbidium species [3,6,8-13].
Ad 169 It is recommended to expand the explanatory / accompanying text of Table 2.
[Answer] As the comment, the title of Table 2 was changed as followed:: Combined genotypes of 12 SSRs in C. goeringii cultivars with samples of 10 or more.
Ad 220-221 Please implement the accompanying text of Table 3.
[Answer] We are sorry to miss a table title. We added a table 3 title: Tracing of miss-matched C. goeringii cultivars (n ≥ 10).
Ad 358 Correctly: …seedlings….
[Answer] As the comment, “seedings” was corrected to “seedlings”.
Scientific Latin names are not highlighted / written in italics in the References chapter several times: 353, 357, 388-389
[Answer] Previously, we just followed the PubMed stile, but we indicated scientific names to italic as your comment.
The use of the English language in the MS corresponds to the quality expected in a scientific paper.
Thank you very much for your valuable comments. The manuscript was edited by an English editing organ (Editage).
Reviewer 4 Report
This manuscript introduced a useful database for the authentication of spring orchid, and generally easy to follow, but still have some minor issues can improve.
1. From the authors' description, all of the samples were collected from mountain, and none of them from the farm or market, thus, it seems not proper to claim that 30% of C. goeringii on the farms and markets may be not genuine;
2. It is not necessary to list all the names of cultivars in the context of 2.1, since it is presented in Table 1;
3. Please check and revise the title of Table 3;
4. Line 278, the author gave a new abbreviation of SG1, which is conflict with the other part of the manuscript, please check and revise it (if necessary).
Author Response
Thank you very much for your valuable comments. We carefully edited the manuscript point by point according to your comments.
From the authors' description, all of the samples were collected from mountain, and none of them from the farm or market, thus, it seems not proper to claim that 30% of C. goeringii on the farms and markets may be not genuine;
[Answer] The cultivars were originally captured from wild mountains, thereafter they were fixed into specific cultivars which are cultivated in the farms.
It is not necessary to list all the names of cultivars in the context of 2.1, since it is presented in Table 1;
[Answer] As the comment, the names of cultivars listed in “2.1. Collection of C. goeringii samples” were deleted, and the paragraph was properly edited.
Please check and revise the title of Table 3;
[Answer] We are sorry to miss a table title. We added a table 3 title: Tracing of miss-matched C. goeringii cultivars (n ≥ 10).
Line 278, the author gave a new abbreviation of SG1, which is conflict with the other part of the manuscript, please check and revise it (if necessary).
[Answer] As the comment, “SG1” was changed to “CG1”.